# Neutrophil Maturation and Survival Is Controlled by IFN-Dependent Regulation of NAMPT Signaling

**DOI:** 10.3390/ijms20225584

**Published:** 2019-11-08

**Authors:** Elena Siakaeva, Ekaterina Pylaeva, Ilona Spyra, Sharareh Bordbari, Benedikt Höing, Cornelius Kürten, Stephan Lang, Jadwiga Jablonska

**Affiliations:** Translational Oncology, Department of Otorhinolaryngology, University Hospital Essen, University Duisburg-Essen, 45147 Essen, Germany; elena.siakaeva@uk-essen.de (E.S.); ekaterina.pylaeva@uk-essen.de (E.P.); ilona.spyra@uk-essen.de (I.S.); sharareh.bordbari@uk-essen.de (S.B.); Benedikt.Hoeing@uk-essen.de (B.H.); Cornelius.Kuerten@uk-essen.de (C.K.); stephan.lang@uk-essen.de (S.L.)

**Keywords:** NAMPT, type I interferons, granulopoiesis, neutrophil progenitors, neutrophil maturation and survival, apoptosis

## Abstract

Granulocyte-colony stimulating factor (G-CSF)/nicotinamide phosphoribosyltransferase (NAMPT) signaling has been shown to be crucial for the modulation of neutrophil development and functionality. As this signaling pathway is significantly suppressed by type I interferons (IFNs), we aimed to study how the regulation of neutrophil differentiation and phenotype is altered in IFN-deficient mice during granulopoiesis. The composition of bone marrow granulocyte progenitors and their *Nampt* expression were assessed in bone marrow of type I IFN receptor knockout (*Ifnar1^-/-^)* mice and compared to wild-type animals. The impact of NAMPT inhibition on the proliferation, survival, and differentiation of murine bone marrow progenitors, as well as of murine 32D and human HL-60 neutrophil-like cell lines, was estimated. The progressive increase of *Nampt* expression during neutrophil progenitor maturation could be observed, and it was more prominent in IFN-deficient animals. Altered composition of bone marrow progenitors in these mice correlated with the dysregulation of apoptosis and altered differentiation of these cells. We observed that NAMPT is vitally important for survival of early progenitors, while at later stages it delays the differentiation of neutrophils, with moderate effect on their survival. This study shows that IFN-deficiency leads to the elevated NAMPT expression in the bone marrow, which in turn modulates neutrophil development and differentiation, even in the absence of tumor-derived stimuli.

## 1. Introduction

Neutrophils are the most frequent leukocytes in human blood. They mature in the bone marrow and are released to the circulation to exert their multiple roles as effector cells in the innate arm of the immune system. These cells are the earliest to arrive at the site of infection, providing the first line of defense against pathogens.

The development of neutrophils in the bone marrow (BM) is tightly regulated. They develop from multipotent hematopoietic stem cells (HSCs) that are contained in the LSK (Lin^−^/Sca-1^+^/c-Kit^+^) cell population. LSK cells are capable of self-renewal and are responsible for producing variable types of mature blood cell lineages. Hematopoietic stem cells undergo progressive commitment to generate multipotent common myeloid progenitor (CMP) cells, which, in turn, can differentiate into either megakaryocyte–erythrocyte progenitor (MEP) or granulocyte–monocyte progenitor (GMP) cells [1]. GMPs terminally differentiate into granulocytes or monocytes. The differentiation pathway from GMPs to functional mature neutrophils contains three neutrophil subsets: a committed proliferative neutrophil precursor (pre neutrophil), which differentiates into non-proliferating immature neutrophils, and later to mature neutrophils [2]. The commitment and differentiation of myeloid progenitors to mature neutrophils are strictly controlled by different factors, the most important being granulocyte-colony stimulating factor (G-CSF), also known as colony-stimulating factor 3 (CSF 3).

It is known that neutrophil differentiation and functionality are modulated by environmental cues, such as G-CSF or type I interferons (IFNs). In this context, nicotinamide phosphoribosyltransferase (NAMPT) is a key molecule involved in G-CSF-triggered granulopoiesis [3]. It is a unique enzyme with cytokine-like features [4]. Intracellularly, NAMPT is a rate-limiting enzyme, converting nicotinamide into nicotinamide adenine dinucleotide (NAD^+^), which in turn activates NAD^+^-dependent protein deacetylases sirtuins that regulate multiple cellular functions [5]. In addition to its enzymatic activity, NAMPT exhibits the cytokine-like functions when released extracellularly [6].

Previously, we showed that type I IFNs play a role in anti-tumor activation of neutrophils. In agreement, IFN-deficient animals displayed an increased amount of immature pro-tumoral neutrophils in tumors. Such tumor-associated neutrophils (TANs) support tumor growth, angiogenesis, and metastasis in mice. Blocking of NAMPT signaling in IFN-deficient mice suppressed this effect [7]. However, mechanisms regulating this process were not well understood. As NAMPT plays an important role during granulopoiesis [3], we hypothesized that NAMPT can be involved in the modulation of neutrophil commitment during their development. Therefore, in this study, we aimed to delineate whether NAMPT was involved in IFN-mediated regulation of neutrophil maturation and survival. Indeed, we could demonstrate that the availability of NAMPT is crucial for the maintenance of early neutrophils progenitor cells as NAMPT inhibition results in accelerated apoptosis of these cells. Inhibition of NAMPT in later stages of neutrophil development, however, leads to their faster maturation and decreased proliferation, especially in IFN-deficient conditions. 

In conclusion, our findings are the first to demonstrate how IFN-dependent NAMPT expression regulates different stages of granulopoiesis.

## 2. Results

### 2.1. Elevated Expression of NAMPT in IFN-Deficient Mice Is Associated with Elevated Survival of Neutrophils 

Being one of the downstream molecules of G-CSF signaling, NAMPT plays a key role in a cell metabolism, increasing survival and tumorigenic properties of different types of cells. To study the regulation of NAMPT by type I IFNs, we performed the analysis of available Gene Expression Omnibus databases GSE65858 [8], Appendix A1). We observed a positive correlation of *NAMPT* with *GCSF* and *G-CSF receptor* (*GCSFR)* gene expression (Figure 1a), and the negative correlation with *interferon receptor (IFNAR)* subunits 1 and 2 in head and neck cancer (HNC) tumor tissue (Figure 1b), demonstrating the enhanced NAMPT expression in the lack of type I IFN signaling.

As neutrophils play a crucial role during tumor development [9,10,11,12], we focused on the regulation of NAMPT signaling and functionality of neutrophils in IFN deficiency. Previously, we could show that tumor growth is accelerated in IFN-deficient mice, as compared to wild type (WT) animals. This was accompanied by the infiltration of such tumors with pro-tumorally biased neutrophils [10,12]. In plasma of tumor-bearing type I IFN receptor knockout (*Ifnar1^-/-^*)mice increased Gcsf levels (Figure 1c) and higher neutrophil counts were observed, as compared to WT animals (Figure 1d). At the same time, no significant differences were observed in the absence of tumors (Figure 1c,d). In tumor tissue, elevated neutrophil infiltration was observed in *Ifnar1^-/-^* mice (Figure 1e). As G-CSF signaling involves NAMPT activation, we have checked the regulation of this factor in neutrophils isolated from IFN-deficient versus WT mice. Indeed, we could observe the upregulation of *Nampt* gene in tumor-associated neutrophils isolated from IFN-deficient mice (Figure 1f). Moreover, IFN-deficient tumor-associated neutrophils showed prolonged survival due to decreased apoptosis (Figure 1g). Functional capacities of such neutrophils were altered, showing elevated tumor-supporting properties (pro-angiogenic, pro-metastatic), with decreased cytotoxicity and immature phenotype (lower CD11b expression).

### 2.2. Different Composition of Bone Marrow Neutrophil Progenitors in Ifnar1^-/-^ Mice Is NAMPT-Dependent and Associated with Altered Apoptosis

While the Gcsf pathway is upregulated in tumor-bearing animals, steady state Gcsf expression is rather low and not significantly altered between *Ifnar1^-/-^* and WT mice. Nevertheless, neutrophils from these mice show also altered functionality. Therefore, we decided to check if tumor-free mice showed any differences concerning neutrophil development and differentiation, which would be associated with IFN-deficiency itself.

The analysis of available Gene Expression Omnibus databases GSE68529 [13] revealed the increase in *Ifnar1* gene expression on short-term HSCs (Lin^-^/cKIT^+^/Sca-1^+^/Flk2^-^/CD48^-^/CD150^-^) as well as on neutrophils on later stages of development, especially immature and mature forms (Mac1^+^/Gr1^hi^) (Figure 1h). Thus, one can expect a major effect of the lack of IFN-signaling on HSC late progenitors and differentiated neutrophils.

We have analyzed the content of neutrophils and their progenitors in BM of IFN-deficient mice and compared it with WT mice (gating strategy is depicted in Figure 2a,b) (adapted from Evrard et al., [2]). Interestingly, we could observe that the number of HSCs (LSK cells; Lin^−^Sca-1^+^c-Kit^+^), is significantly decreased in IFN-deficient mice. The same was true for bipotent GMP cells (single viable Lin^-^/Sca-1^-^/c-KIT^+^/CD34^+^/CD16/32^+^) and proliferating pre-neutrophils (single viable CD3e^-^/CD45R^-^/NK-1.1^-^/CD11b^+^/c-KIT^+^/CXCR4^+^) (Figure 2c). The levels of immature (single viable CD3e^-^/CD45R^-^/NK-1.1^-^/CD11b^+^/c-KIT^-^/CXCR4^-^/Ly6G^+^/CXCR2^-^) and mature (single viable CD3e^-^/CD45R^-^/NK-1.1^-^/CD11b^+^/c-KIT^-^/CXCR4^-^/Ly6G^+^/CXCR2^+^) neutrophils in bone marrow increased in IFN-deficient mice (Figure 2d).

The analysis of available Gene Expression Omnibus databases GSE68529 [13] showed a progressive increase of *Nampt* expression during maturation, with the highest levels in immature and mature forms (Mac1^+^/Gr1^hi^) (Figure 2e). We verified these findings in our own cohort of mice using qRT-PCR. Importantly, IFN-deficiency was associated with more prominent elevation of *Nampt* expression in mature neutrophils (Figure 2f), which complements the published data about up-regulation of *Nampt* in IFN deficiency in circulating and tumor-associated neutrophils [7].

The observed dysregulation in populations of neutrophil progenitors can be due to NAMPT-related disbalance of proliferation and apoptosis of such cells. To prove the effect of NAMPT on these functions, we used the specific inhibitor of NAMPT (FK866). While no differences in proliferation rate between *Ifnar1^-/-^* and WT neutrophil progenitors was observed (Figure A1a,b), we focused the attention on the regulation of survival and differentiation of neutrophil progenitors by NAMPT. We were able to show that apoptosis was higher in *Ifnar1^-/-^* mice in early stages of granulopoiesis, but significantly lower in mature IFN-deficient neutrophils in bone marrow (Figure 3a,b).

As increased expression of *Casp3* gene is known to sensitize cells to apoptosis induced by various stimuli [14], we checked its expression in neutrophil progenitors. In line with decreased survival of *Ifnar1-/-* pre-neutrophils, the expression of *Casp3* was higher in these cells, as compared to WT. The opposite tendency was observed in mature neutrophils (Figure 3c). Importantly, the negative correlation between *Nampt* and *Casp3* in mature neutrophils (*R* = −0.57, *p* = 0.05), but not in pre-neutrophils and immature cells (*R* = 0.03, *p* = 0.91, and *R* = 0.47, *p* = 0.12, respectively), was observed, suggesting the role of NAMPT up-regulation in the prevention of apoptosis of mature neutrophils.

### 2.3. Direct NAMPT Inhibition in Isolated BM Progenitors Regulates Their Survival and Differentiation, Depending on the Maturation Stage

To confirm the role of NAMPT during granulopoiesis, we decided to recapitulate development of neutrophils from progenitors as described by Gupta et al., 2014 [15]. Shortly, isolated BM progenitor cells were differentiated in vitro into granulocyte progenitors and mature forms after addition of stem cell factor (SCF) and Interleukin 3 IL-3, followed by G-CSF supplementation. We evaluated the effects of early (at day 0, before stimulation) and late (at day 5) NAMPT inhibition in this experimental setting. The scheme of the experiment is shown in the Figure 4a.

Being applied at the very early steps of differentiation, FK866 induced apoptosis and death of the majority of cells (Figure 4b). To the opposite, FK866 treatment of the cells after 5 days of differentiation in culture resulted in only a moderate increase of apoptosis (Figure 4c), accompanied by enhanced differentiation into immature and mature neutrophils (Figure 4d,e). Moreover, while the absence of IFN signaling was associated with lower levels of CD11b on progenitors and mature neutrophils in BM, inhibition of NAMPT in *Ifnar1^-/-^* led to an increase of CD11b expression of the cells (Figure 4f), suggesting the induction of more mature phenotype of neutrophils after NAMPT block.

This all shows that NAMPT signaling is vitally important for early stages of neutrophil progenitors, while at the later stages it delays cell maturation.

### 2.4. NAMPT Is Responsible for Survival and Delayed Differentiation of Murine Myeloblast Cell Line

To prove the mechanism of FK866-mediated alteration of neutrophil differentiation and maturation, we used a model of neutrophil progenitors, the murine myeloblast cell line 32D. This cell line consists of immature neutrophils that can be in vitro differentiated into neutrophils using G-CSF, similarly to neutrophil progenitors in the bone marrow [16,17]. We treated such progenitors with FK866 at different time points of their development (Figure 5a). Of note, we could observe that immature neutrophils that were treated with FK866 early in their development seem to be particularly vulnerable, as they died upon treatment (Figure 5b). Treatment of neutrophils at later stages of differentiation results only in moderate decrease of their viability (Figure 5b,c) and it mainly supports their maturation. Thus, elevated percentage of CD11b^+^ neutrophils and their enhanced CD11b expression is observed, compared to untreated controls (Figure 5d). Such cells show more mature phenotype characterized with segmented nuclei (Figure 5e,f).

### 2.5. Inhibition of NAMPT in Human Neutrophil Progenitors Reduces Their Survival and Improved Differentiation

Since NAMPT pathway seems to regulate neutrophil differentiation and survival in the mouse system, we decided to validate our observations in human neutrophils. To provide the long term study mimicking the differentiation of neutrophil progenitors in bone marrow, we used human HL-60 cells that are widely used as neutrophil equivalent for molecular manipulation [18,19]. The scheme of the experiment and gating strategy are shown in Figure 6a,b. Importantly, we observed the increasing *Nampt* expression during DMSO-G-CSF-induced maturation of the cells (Figure 6c), similarly to the murine system (Figure 4e,f). FK866 treatment of immature HL-60 led to immediate death of these cells, as their survival was probably strongly dependent on the NAMPT pathway. To the contrary, treatment of differentiated HL-60 neutrophils affected their viability only slightly (Figure 6d), but mainly induced their maturation. CD71, a marker of immature cells, was reduced on these cells, and the ratio between CD11b^+^ (mature) and CD71^+^ (immature) neutrophils was significantly increased (Figure 6e,f). Giemsa staining confirmed increased percentage of band granulocytes in the sample treated with FK866 on day 3 of maturation (Figure 6g). 

To confirm these data in the primary human neutrophils, we treated mature neutrophils isolated from blood of healthy individuals with G-CSF or FK866. We could observe a decreased apoptosis rate upon G-CSF treatment, and similarly to cell line data, increased apoptosis after blockage of NAMPT signaling (Figure 6h). 

Together, these results suggest that similar neutrophil differentiation mechanisms exist in mice and human. In both organisms neutrophil maturation and survival depend on the activity of NAMPT signaling pathway. NAMPT is vitally important for survival of early progenitors, while at later stages it delays the differentiation of cells with moderate effect on their survival. This may explain the changes in bone marrow composition in type I IFN deficiency, which correlates with the upregulation of NAMPT signaling, even in the absence of tumor-derived stimuli. Inhibition of NAMPT at late stages of neutrophil development improves their differentiation and leads to more mature phenotype.

## 3. Discussion

Neutrophils constitute the first line of immune defense against pathogens, but also play an essential role in the regulation of tumor progression. We can observe that these cells could support or inhibit tumor development, depending on their activation status [9,10,11,12,20]. The mechanism responsible for this phenomenon is not clear, but NAMPT seems to be involved in this process [7].

NAMPT, as a part of G-CSF signaling, regulates numerous cellular functions through the protein deacetylation activity of NAD^+^-dependent sirtuins. Sirtuins activate cEBPα and β levels that in turn stimulate synthesis of G-CSF and its receptor G-CSFR. This results in the induction of granulopoiesis [3]. Our findings prove the correlation between *NAMPT* and *GCSF* gene expression in human tissues. Interestingly, G-CSF is known to be a key growth factor for neutrophils. It has been shown that during inflammation, such as in tumor development, increased amounts of G-CSF potentiate neutrophil maturation and their mobilization from the bone marrow [21,22]. Neutrophil progenitors expand under tumoral stress and immature neutrophils are recruited to the periphery of tumor-bearing mice [2].

Here, we show that G-CSF levels are elevated during tumorigenesis, leading to the massive mobilization of pro-tumoral immature neutrophils in mice, which is especially apparent in IFN-deficient mice [20,23]. In line with these findings, we observed that type I IFNs suppress G-CSF/ NAMPT signaling, resulting in the reduced neutrophil content in blood and tumors, and in suppressed tumor development [7]. Here, we proved the negative correlation between the expression of IFNAR subunits and NAMPT, supporting the hypothesis about down-regulation of NAMPT by type I IFNs. While in the absence of tumor the levels of G-CSF are low, we were interested to check the hypothesis whether the lack of type I IFNs still induce NAMPT-mediated changes in neutrophil development and differentiation, resulting in altered activation of these cells already in bone marrow.

The analysis of available Gene Expression Omnibus database revealed two peaks of increased *Ifnar1* on bone marrow cells, in short-term HSC and in late progenitors and mature neutrophils, suggesting the most vulnerable time points of neutrophil development that are influenced by IFN deficiency. Previous studies showed that type I IFNs activate dormant stem cells [24] and increase the HSC proliferation [25]. Buechler at al. have shown that type I IFNs activate myeloid development of CMPs in vitro by the toll-like receptor (TLR)-associated mechanism and promote monocyte/macrophage development from progenitors [26]. Our experiments suggest that type I IFNs are involved in both, in the very early stage of granulopoiesis and during final maturation of neutrophils. Comparing neutrophil progenitors between bone marrow of WT and *Ifnar1^-/-^* mice, we could observe the most pronounced differences in the compartment of HSCs and mature neutrophils. A significant decrease of LSK cells in IFN-deficient mice was observed, which is in line with Smith et al. [25]. In agreement with our data, other studies proved the strong influence of type I IFNs on mature circulating and tumor-associated neutrophils, leading to their improved survival [10,12,20]. 

NAMPT has been postulated to have an essential role in myelopoiesis as it was shown to induce granulocytic differentiation of CD34^+^ hematopoietic progenitor cells [3]. Moreover, several studies have shown that NAMPT, in its non-enzymatic state, can act as a growth factor [27]. Until now, it has been shown that NAMPT stimulates growth of endothelial cells, tumor cells and B cells [28,29]. Our results shows that the highest NAMPT expression is observed in late stages of granulopoiesis, in mature neutrophils, and this is augmented by the lack of IFNs. NAMPT is essential for replenishment of intracellular NAD^+^ pool [30]. As NAD^+^ is needed in many critical cellular processes, such as transcription, cell-cycle progression, DNA repair, and regulation of apoptosis [31,32], maintaining NAD^+^ content in cells prevents their death [33]. Moreover, NAMPT was suggested to activate G1-S cell cycle progression, thus promoting cell survival [34]. In agreement, activation of pro-survival cascade by NAMPT has been demonstrated in tumoral [35] and non-tumoral cells [36]. NAMPT has been shown to protect monocyte lineage from endoplasmic-reticulum (ER) stress-induced apoptosis [6] and to downregulate Caspases 3, 8, and 9 in neutrophils [37]. Overexpression of NAMPT has been demonstrated to delay senescence and substantially lengthened life span of smooth muscles cells. Moreover, it enhanced their resistance to oxidative stress [38]. Similarly, stable NAMPT-overexpressing cells were shown to be more resistant to apoptosis [33], possibly due poly(ADP-ribose)polymerase (PARP) overexpression [39]. It has been shown that NAMPT can delay cellular senescence [30]. The anti-aging phenomenon of NAMPT is possibly mediated by enhanced sirtuin1 deacetylase activity that, in turn, holds p53 levels below those, which induce senescence [40]. 

Another mechanism of NAMPT-related survival is the activation of mammalian target of rapamycin (mTOR) signaling, as FK866 treatment significantly downregulated mTOR in hepatocarcinoma cells [41]. Moreover, it has been shown that mTOR deletion in myeloid cells promotes eosinophil development, but suppresses neutrophil development [42]. As mTOR promotes the differentiation of adult stem cells, driving their growth and proliferation [43], the inhibition of this pathway leads to their rapid apoptosis. Importantly, our results show, that IFN-deficiency was associated with strong up-regulation of NAMPT in mature neutrophils in BM even in steady state. The survival of such IFN-deficient cells was significantly elevated due to NAMPT-related decrease in *Casp3* expression, as compared to WT. 

NAMPT is involved in proliferative signaling pathways [44], such as PI3K-AKT-mTOR signaling. These pathways has been shown to be essential for maintenance of functional hematopoietic stem cells (HSCs) [45]. Elevated mTOR signaling drives and supports the hyperproliferation of HSCs [43], possibly due to enhanced reactive oxygen species (ROS) production [46]. This contributes to HSC depletion and the expansion of later progenitor cells [47]. 

NAMPT provides important NAD^+^ pool in proliferating progenitor cells that is crucial for their proliferation. Mechanistically, FK866 induces cell death through NAD^+^ depletion that leads to ATP depletion and finally to the cell death [30]. As proliferating cells need more NAD^+^, such cells are more vulnerable to NAMPT inhibition [30]. This would explain observed potent anti-tumor activity of FK866 in hematological malignancies [48]. In agreement, it has been demonstrated that FK866 induces apoptosis and reduces proliferation of leukemic cells by depletion of intracellular NAD^+^. The metabolic rate of fast proliferating cells, such as progenitors or cancer cells, is abnormally high; therefore, they require higher NAD^+^ levels. Because of this, cancer cells have an increased sensitivity toward low levels of the cellular NAD^+^ content and are more susceptible to NAMPT inhibition than normal cells [40]. The overall effect of FK866 treatment on the early neutrophil progenitors is a sum of its pro-apoptotic and anti-proliferative effects. Thus, remaining cells are not able to restore the progenitor pool. 

Extracellular NAMPT has been shown to bind to TLR4 expressed by multipotent hematopoietic stem cells [49]. This triggers cell cycle entry of these cells through the activation of NFκB [50,51]. Moreover, activation of TLR4 upregulates Sca-1 expression on such precursors, that supports the development of granulocyte lineage [52,53]. Therefore, early neutrophil progenitors seem to be particular vulnerable for NAMPT depletion. 

Previously, we have observed that NAMPT signaling promotes pro-tumoral bias of tumor-associated neutrophils [7]. In agreement, other groups also observed NAMPT-mediated myeloid cell polarization. As example, Grolla et al. observed elevated M1 polarization of macrophages in response to NAMPT treatment [44], while others observed rather M2 polarization in these conditions [34]. While the reasons for these conflicting results has to be still elucidated, the modulatory effect of NAMPT signaling on the differentiation of myeloid cells is unquestionable. Possibly, modulatory effect of NAMPT in the polarization and activation of myeloid cells depends on their maturation stage. 

In IFN-deficiency, we observe an NAMPT-related delay in the maturation of neutrophils. Selective inhibition of NAMPT using FK866 resulted in maturation and higher CD11b expression on neutrophils. Importantly, similar results were obtained in murine and human neutrophil cell lines, suggesting similar regulation of neutrophil development by NAMPT in different species.

Treatment of neutrophil progenitors with FK866 revealed diverse susceptibility to NAMPT depletion in different progenitor stages. While early progenitors undergo fast apoptosis upon treatment, suggesting their strong dependence on the NAMPT pathway, later progenitors rather undergo an accelerated maturation. This switch in the response to NAMPT depletion mirrors differently activated signaling pathways at different granulopoiesis stages. NAMPT inhibition in early progenitors expressing low levels of NAMPT is critical for cell survival. However, depletion at later stages of development, when cells express its elevated amounts, is not as detrimental for neutrophil survival, as these cells decrease already their proliferation rate and do not need as high amounts of NAD+. Moreover, it is possible that due to high levels of NAMPT in such cells the inhibition is incomplete and they can fast recover their NAMPT pool. Taken together, NAMPT controls multiple stages of neutrophil maturation and survival, and IFN availability regulates these processes. 

## 4. Materials and Methods

### 4.1. Animals

Mice (8–12 weeks old) of C57BL/6J wild-type (WT) and *Ifnar1^-/-^* strains were used in all experiments. Mice were bred and kept under specific pathogen free (SPF) conditions in the animal facility of the University Hospital Essen (Essen, Germany). 

### 4.2. Tumor Model and Tumor-Associated Neutrophils

B16F10 melanoma cells were purchased from DSMZ Braunschweig, Germany and cultivated in IMDMc (IMDM, Iscove’s Modified Dulbecco’s medium) (Gibco, Thermo Fisher Scientific, Waltham, MA, USA) supplemented with 10% FCS (Biochrom, Merck, Darmstadt, Germany) and 1% penicillin/streptomycin (Gibco, Thermo Fisher Scientific, Waltham, MA, USA) at 37 °C and 5% CO_2_. The cell line was regularly tested for mycoplasma contamination using the Venor GeM Kit (Minerva Biolabs, Berlin, Germany) and was negative. Tumor cells were injected subcutaneously into C57BL/6J (WT) and *Ifnar1^-/^*^-^ mice, and tumor growth was monitored. After 14 days, mice were sacrificed. Heparinized blood was collected via heart puncture and plasma was prepared after centrifugation. Tumors were harvested and digested using dispase II 0.2 µg/mL, collagenase D 0.2 µg/mL, and DNase I 100 µg/mL (all from Sigma-Aldrich, Merck KGaA, St. Louis, MO, USA) solution in DMEMc (DMEM (Dulbecco’s Modified Eagle Medium) Gibco, Thermo Fisher Scientific, Waltham, MA, USA), 10% FCS, 1% penicillin/streptomycin). Cells were meshed through 50-µm filters (Cell Trics, Sysmex, Goerlitz, Germany) and erythrocytes lysed in ACK buffer. Single-cell suspensions were stained with antibodies and viability dyes listed below. 

### 4.3. Bone Marrow Cells Isolation

Bone marrow (BM) was isolated from femurs and tibias via crushing. The erythrocytes were lysed in ACK buffer containing NH_4_Cl 150 mM, KHCO_3_ 10 mM, Na_2_EDTA 0.1 mM. pH 7.2. Single-cell suspensions were meshed through 50-µm filters (Cell Trics, Sysmex, Goerlitz, Germany) and stained with antibodies and viability dyes listed below or used for in vitro maturation assay.

### 4.4. Murine Neutrophil Isolation

The following neutrophil subpopulations: viable CD11b^+^/c-KIT^+^/CXCR4^+^ pre-neutrophils, viable CD11b^+^/c-KIT^-^/CXCR4^-^/Ly6G^+^/CXCR2^-^ immature neutrophils, and viable CD11b^+^/c-KIT^-^/CXCR4^-^/Ly6G^+^/CXCR2^+^ mature neutrophils were sorted using a FACS Aria cell sorter (BD Biosciences, Franklin Lakes, NJ, USA) and the purity of cells was assessed (≥ 90%) (Figure A2).

### 4.5. In Vitro Maturation of Isolated Murine Bone Marrow Progenitors

Murine BM progenitor cells were negative selected by depletion of CD3e^+^CD45R^+^NK1.1^+^CD11b^+^Ter119^+^ BM cells using the Streptavidin MicroBeads and LD Columns from Miltenyi Biotec (Bergisch Gladbach, Germany) according to the manufacturer’s protocols. For maturation isolated cells were cultured in 12-well plates at the concentration of 0.3 × 10^6^ cells/mL in IMDMc for 7 days at 37 °C and 5% CO_2_. At day 0, the cells were stimulated with murine (m) SCF (mSCF) and mIL-3, at day 3 after medium change with mSCF, mIL3, and mG-CSF and at day 5 also after medium change only with mG-CSF (all from Peprotec, Hamburg, Germany; end concentration 50 ng/mL) (according to the protocol described by D. Gupta et al., 2015, adapted [15]). The cells were also treated with FK866 from day 0 or from day 5.

### 4.6. In Vitro Proliferation of Isolated Murine Bone Marrow Progenitors

After 7 days in culture cells were permeabilized with ice-cold 70% ethanol and stained with KI-67 antibody (see below).

### 4.7. Cell Lines

The murine myeloid cell line 32D (C3H) is one of the few cell lines that can terminally differentiate into neutrophils. It is widely used as a model for in vitro studies of neutrophil differentiation and activation. In 32D the neutrophilic differentiation is induced using G-CSF to an extent that mimics very closely the differentiation pattern that occurs in the normal bone marrow in vivo [16,17]. 32D was purchased from DSMZ Braunschweig, Germany. 

Human HL-60 cells consist predominantly (greater than 90%) of promyelocytes [18]. These cells can be used as a neutrophil equivalent for molecular manipulation studies [18,19]. HL-60 cells acquire a neutrophilic phenotype after DMSO treatment [54]; addition to the medium G-CSF enhances their maturation [55]. HL-60 was purchased from DSMZ Braunschweig, Germany.

32D neutrophils were cultivated in IMDMc and HL-60 neutrophils in RPMIc medium (RPMI-1640 (Gibco, Thermo Fisher Scientific, Waltham, MA, USA), 10% FCS and 1% penicillin/streptomycin) at 37 °C and 5% CO_2_. All cell lines were tested for mycoplasma contamination and were negative.

In vitro treatment of the 32D cell line FK866 (100 nM) was added at the beginning of G-CSF (10 ng/mL) induced maturation (day 0), on day 3, 6 or 9. On day 14 cells were harvested for analysis (Figure 5a).

In vitro treatment of HL-60 cell line: FK866 (100nM) was added to the cells before beginning of DMSO (1,5% v/v) and G-CSF (10 ng/mL) induced maturation, at the beginning (day 0) or at day 3. On day 7 cells were harvested for analysis (Figure 6a).

### 4.8. Morphological Analysis

For morphological analysis, cytospin preparations of 32D and HL-60 cells on SuperfrostTM slides (Gibco, Thermo Fisher Scientific, Waltham, MA, USA) were fixed with pure methanol, stained by Giemsa (Sigma-Aldrich, Merck KGaA, St. Louis, MO, USA) and nuclear morphology was assessed using light microscope Olympus BX5 (Olympus, Tokyo, Japan). At least 10 fields of view were counted and percentages of immature, band, and segmented nuclei were calculated.

### 4.9. Isolation and Treatment of Human Peripheral Blood Neutrophils

Peripheral blood obtained from healthy donors was drawn into 3.8% sodium citrate anticoagulant monovettes (Sarstedt, Nuembrecht, Germany) and mixed 1:1 with PBS (Gibco, Thermo Fisher Scientific, Waltham, MA, USA) before separation by density gradient centrifugation (Biocoll density 1077 g/mL, Biochrom, Merck, Darmstadt, Germany). The mononuclear cell fraction was discarded and neutrophils (purity ≥ 95%, Figure A3) were isolated by sedimentation over 1% polyvinyl alcohol, followed by hypotonic lysis (0.2% NaCl) of erythrocytes.

In vitro treatment of isolated neutrophils was performed with G-CSF (PeproTech, Hamburg, Germany) 10 ng/mL and FK866 (100 nM) for 18 hours at 37 °C and 5% CO_2_.

In view of the emerging diversity of circulating neutrophil subtypes in humans it has to be noted that high-density neutrophils, not low-density neutrophils, have been investigated in this study.

### 4.10. Apoptosis Assay

Spontaneous apoptosis of murine BM neutrophils, 32D and HL-60 cell lines, and isolated human blood neutrophils was determined by Annexin V/7-aminoactinomycin (7AAD) apoptosis detection kit (BD Biosciences, Franklin Lakes, NJ, USA) after an 18 hours of incubation in DMEMc or RPMIc medium, respectively, with stimuli and inhibitors. The percentages of alive (annexin V^–^ /7-AAD^–^), apoptotic (early apoptosis, annexin V^+^ /7-AAD^–^), and dead by apoptosis (late apoptosis, annexin V^+^ /7-AAD ^+^) cells from all single cells were estimated.

### 4.11. Flow Cytometry

The samples were assessed using BD FACS Canto II system and data were analyzed using BD FACS Diva software (BD, Franklin Lakes, NJ, USA).

### 4.12. Antibodies

Anti-mouse ABs: anti-CD3e (clone: 145-2C11), anti-CD34 (clone: HM34), anti-CD45R (clone: RA3-6B2), anti-CD117 (c-KIT, clone: 2B8), anti-CD182 (CXCR2, clone: SA044G4), anti-CD184 (CXCR4, clone: L276F12), anti-Ly-6A/E (Sca-1, clone: D7), anti-Ki-67 (clone: 11F6), anti-Ly-6G (clone: 1A8), anti-NK-1.1 (clone: PK1), anti-mouse/human-CD11b (clone: M1/70) all from BioLegend (San Diego, CA, USA), and anti-CD16/32 (clone: 93) and anti-CD62L (clone: MEL-14) (both from eBioscience, Thermo Fisher Scientific, Waltham, MA, USA). Anti-human ABs: anti-CD66b (clone: 80H3, Beckman Coulter, Brea, CA, USA), and anti-CD71 (clone: M-A712, BD Biosciences, Franklin Lakes, NJ, USA). Viability dye eFluor780 and eFluor506 (eBioscience, Thermo Fisher Scientific, Waltham, MA, USA) or DAPI (BioLegend, San Diego, CA, USA) were used to determine viable cells.

### 4.13. Quantitative Real Time Polymerase Chain Reaction (qRT-PCR)

The RNA was isolated using Qia Shredder and RNeasy Mini Kit (Qiagen, Hilden, Germany) and the cDNA was produced using the Superscript II Reverse Transcriptase Kit (Invitrogen, Thermo Fisher Scientific, Waltham, MA, USA). qRT-PCR was performed at 60 °C annealing temperature using primers listed below. As housekeeping genes, *Rps9* was used for murine samples and *BACTIN* for human samples. The mRNA expression was measured using the Luna Universal qPCR Master Mix (New England BioLabs, Ipswich, MA, USA).

Murine primers: *Rps9*-f: 5’-TTGACGCTAGACGAGAAGGAT-3’, *Rps9*-r: 5’-AATCCAGCTTCATCTTGCCCT-3’, *Caspase3*-f: 5’-ATGGGAGCAAGTCAGTGGAC-3’, *Caspase3*-r: 5’-TTGAGGTAGCTGCACTGTGG-3’, *Nampt*-f: 5‘-TACAGTGGCCACAAATTCCA-3‘, *Nampt*-r: 5‘-CAATTCCCGCCACAGTATCT-3‘. Human primers: *BACTIN*-f: 5’-AGCGGGAAATCGTGCGTG-3’, *BACTIN*-r: 5’-GGGTACATGGTGGTGCCG-3’, *NAMPT*-f: 5’-GCAGAAGCCGAGTTCAACATC-3’, *NAMPT*-r: 5’-TGCTTGTGTTGGGTGGATATTG-3’.

### 4.14. G-CSF Measurements

G-CSF content in plasma was analyzed with ELISA (R&D Systems, Minneapolis, MN, USA) according to the manufacturer’s protocols.

### 4.15. Analysis of the Data Deposited in the Gene Expression Omnibus Databases 

We used previously published microarray data deposited in the Gene Expression Omnibus databases GSE65858 [8] and GSE68529 [13]. Analyzed samples are listed in Appendix A1 and 2, respectively.

### 4.16. Statistics

Statistical analyses were performed using Kruskal-Wallis ANOVA for multiple comparisons with the Bonferroni correction, Mann–Whitney U-test for two independent samples and the Wilcoxon test for dependent samples. Correlations were estimated with Spearman rank R test. *p* < 0.05 was considered significant.

## 5. Conclusions

In sum, we have identified an important association between IFN-dependent NAMPT expression and the regulation of neutrophil maturation in mice and human. We were able to demonstrate that the progressive increase of NAMPT expression during neutrophil progenitor maturation in the bone marrow is suppressed by type I IFNs. Changed composition of bone marrow progenitors in WT versus IFN-deficient mice correlates with the dysregulation of apoptosis and altered differentiation of neutrophils. We show that NAMPT is vitally important for survival of early neutrophil progenitors, while at later stages it mainly delays the differentiation of these cells with a moderate effect on their survival. This is responsible for the different functionality of neutrophils in IFN-deficient and IFN-sufficient mice.

## Figures and Tables

**Figure 1 ijms-20-05584-f001:**
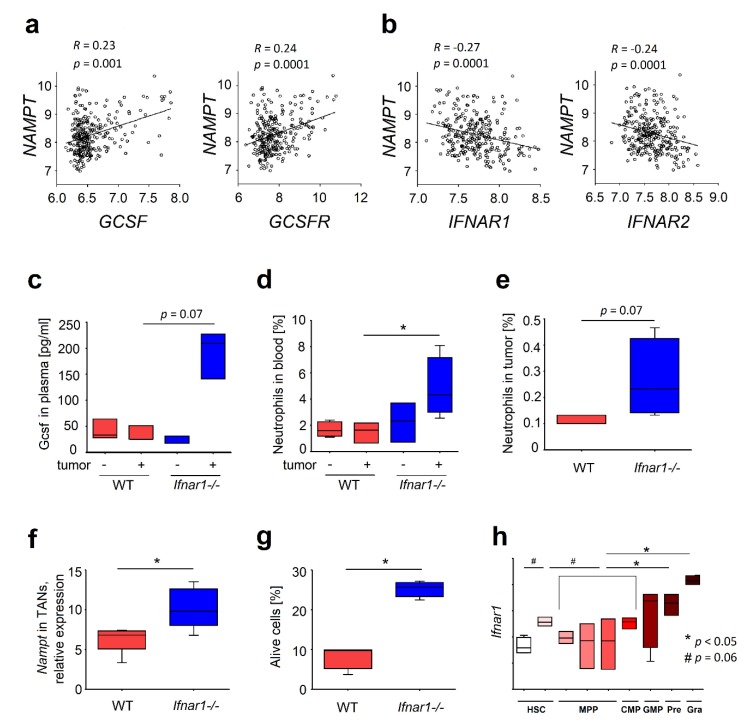
Expression of the granulocyte-colony stimulating factor (G-CSF) pathway and neutrophil infiltration depends on type I interferon availability during tumor development. (**a**) Positive correlation between *nicotinamide phosphoribosyltransferase (NAMPT)* and *G-CSF* genes and *NAMPT* and *G-CSF-receptor (GCSFR)* gene expression in the whole tumor cells population of head-and-neck cancer patients; (**b**) Negative correlation between *NAMPT* gene and interferon receptor (*IFNAR)* subunit 1 and 2 gene expression (*IFNAR1, IFNAR2*, respectively) in tumor tissue of head-and-neck cancer patients. *n* = 270; (**c**) Up-regulated level of Gcsf in plasma of type I IFN receptor knockout (*Ifnar1^-/-^)* tumor-bearing mice in comparison to wild-type (WT) mice; (**d**) Elevated percentage of neutrophils in blood and (**e**) tumor of tumor-bearing *Ifnar1^-/-^* mice in comparison to WT mice; (**f**) Elevated *Nampt* gene expression in tumor-associated neutrophils (TANs) of *Ifnar1^-/-^* mice in comparison to WT mice; (**g**) Survival of TANs in increased in *Ifnar1^-/-^* mice; (**h**) Expression of *Ifnar1* gene in different stages of granulopoiesis in WT mice. *n* = 3–4 mice per group. Data are presented as median, 25, 75 percentiles and minimal and maximal values. * *p* < 0.05. NAMPT: nicotinamide phosphoribosyltransferase.

**Figure 2 ijms-20-05584-f002:**
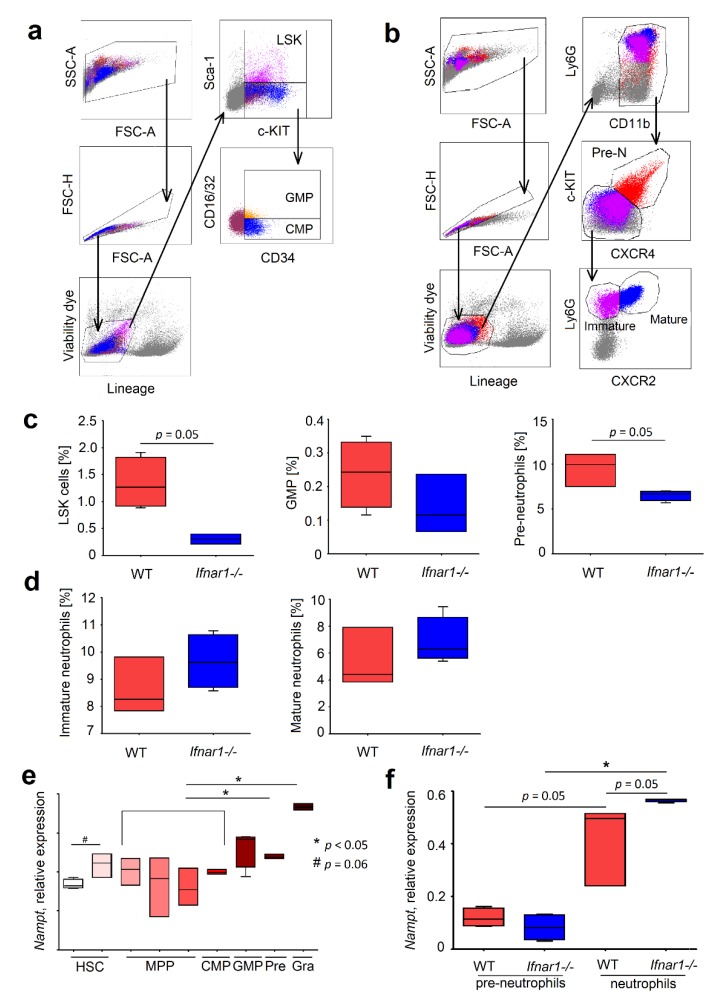
Interferon (IFN) availability influences *Nampt* expression in neutrophil progenitors and the course of granulopoiesis in mice. (**a**) Gating strategy of LSK cells (single viable Lin^−^/Sca-1^+^/c-KIT^+^), common myeloid progenitors (CMP) (single viable Lin^-^/Sca-1^-^/c-KIT^+^/CD34^+^/CD16/32^-^), and granulocyte–monocyte progenitors (GMPs) (single viable Lin^-^/Sca-1^-^/c-KIT^+^/CD34^+^/CD16/32^+^); (**b**) Gating strategy of pre-neutrophils (single viable CD3e^-^/CD45R^-^/NK-1.1^-^/CD11b^+^/c-KIT^+^/CXCR4^+^ cells), immature neutrophils (single viable CD3e^-^/CD45R^-^/NK-1.1^-^/CD11b^+^/c-KIT^-^/CXCR4^-^/Ly6G^+^/CXCR2^-^ cells), and mature neutrophils (single viable CD3e^-^/CD45R^-^/NK-1.1^-^/CD11b^+^/c-KIT^-^/CXCR4^-^/Ly6G^+^/CXCR2^+^ cells); (**c,d**) Neutrophil progenitors in different stages of differentiation presented as a percentage from all viable cells from bone marrow of WT and *Ifnar1^-/-^ mice*; (**e**) Expression of *Nampt* gene in neutrophil progenitors in different stages of maturation; (**f**) Expression of *Nampt* gene in pre- neutrophils and neutrophils from bone marrow of WT and *Ifnar1^-/-^* mice. *n* = 3–4 mice per group. Data are presented as median, 25th and 75th percentiles, and minimal and maximal values. * *p* < 0.05.

**Figure 3 ijms-20-05584-f003:**
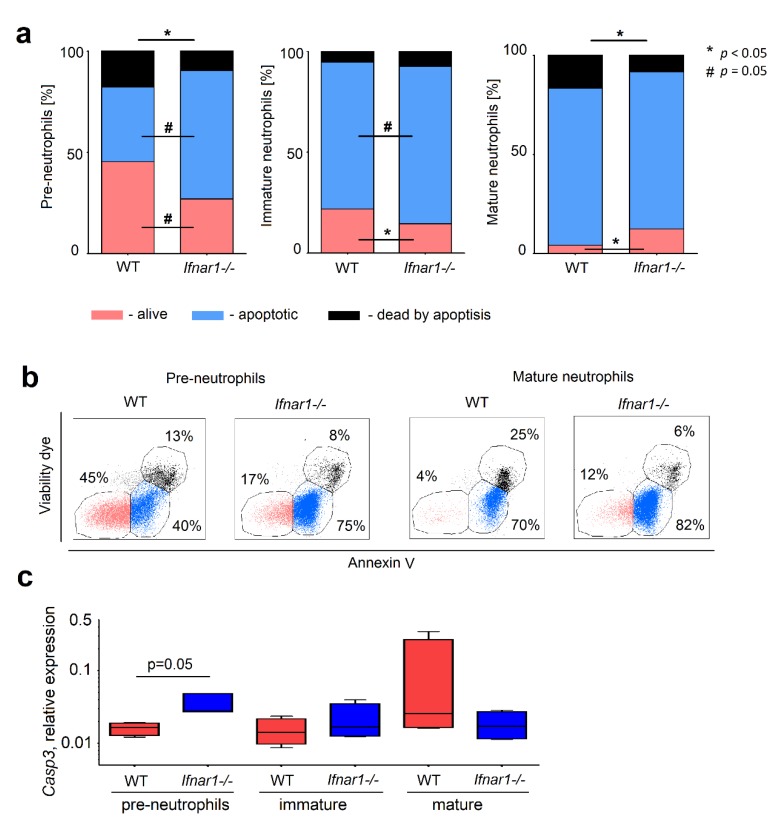
IFN-deficiency is associated with decreased survival of early progenitors, but increased survival of mature neutrophils in bone marrow. (**a**) Percentage of alive (annexin V^–^/7-aminoactinomycin (7-AAD^–^)), apoptotic (early apoptosis, annexin V^+^/7-AAD^–^), and dead by apoptosis (late apoptosis, annexin V^+^/7-AAD^+^) cells from all single cells in late stages of neutrophils maturation. Pre-neutrophils, immature and mature neutrophils from bone marrow of naive WT and *Ifnar1^-/-^* mice were isolated after 18 h of staining with annexin V and viability dye 7-AAD; (**b**) Gating strategy of alive, apoptotic, and dead by apoptosis cells and examples of populations for pre- and mature sorted neutrophils of WT and *Ifnar1^-/-^* mice; (**c**) Relative expression of *Casp3* gene in pre-, immature and mature neutrophils. *n* = 4 mice per group. Data are presented as median, 25th and 75th percentiles, and minimal and maximal values. * *p* < 0.05.

**Figure 4 ijms-20-05584-f004:**
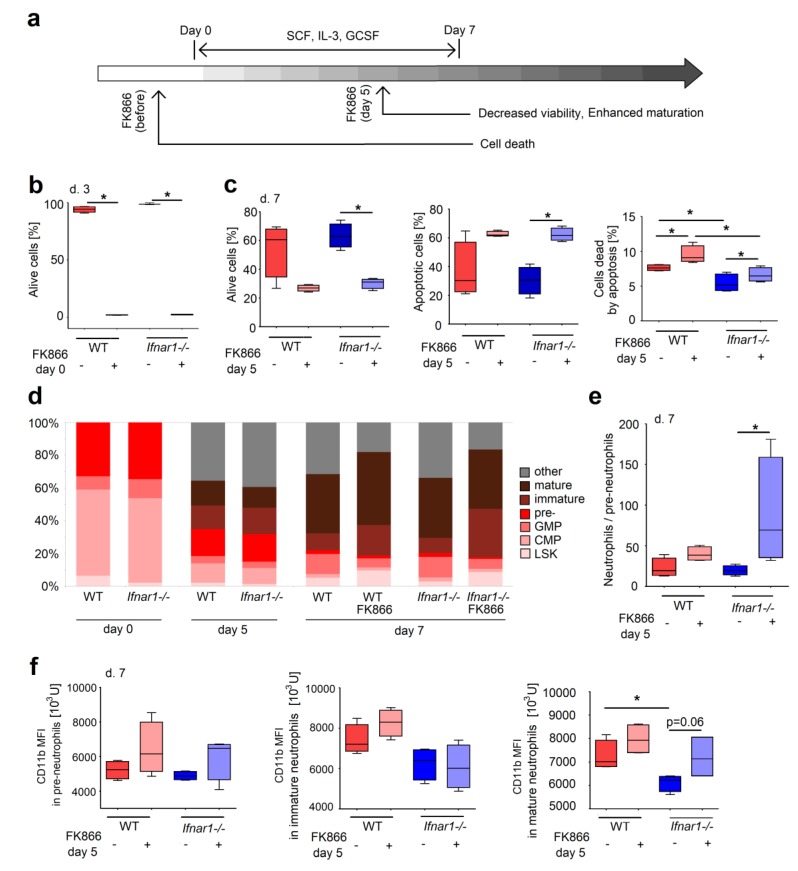
NAMPT inhibition results in apoptosis of early progenitors and maturation of differentiated neutrophils. (**a**) The scheme of the experiment. Progenitor cells were isolated from bone marrow of wild-type (WT) and *Ifnar1^-/-^* mice. Maturation was induced by stem cell factor (SCF) and IL-3, followed by G-CSF supplementation. Treatment with FK866 was performed at days 0 and 5; (**b**) FK866 treatment of isolated progenitors led to cell death. Cells were stained with annexin V and viability dye 7-AAD. Data are presented as a percentage of alive (annexin V^–^/7-AAD^–^) cells from all single cells; (**c**) FK866 treatment in late stages of neutrophils differentiation increases apoptosis in cells from WT mice in comparison to cells from *Ifnar1^-/-^* mice. Data are presented as a percentage of alive (annexin V^–^ /7-AAD^–^), apoptotic (early apoptosis, annexin V^+^/7-AAD^–^), and dead by apoptosis (late apoptosis, Annexin V^+^/7-AAD^+^) cells from all single cells; (**d**) The relative amount of neutrophils progenitors in different stages of maturation of cells from WT and *Ifnar1^-/-^* mice; (**e**) FK866 treatment in late stages of neutrophils differentiation increases the ratio of neutrophils (immature + mature) in relation to pre-neutrophils in *Ifnar1^-/-^* mice; (**f**) CD11b expression on pre-, immature and mature neutrophils differentiated from progenitors from WT and *Ifnar1^-/-^* mice. FK866 treatment in late stages of neutrophils differentiation led to increased CD11b expression on mature neutrophils from *Ifnar1^-/-^* mice. *n* = 4. Data are presented as median, 25th and 75th percentiles, and minimal and maximal values. * *p* < 0.05.

**Figure 5 ijms-20-05584-f005:**
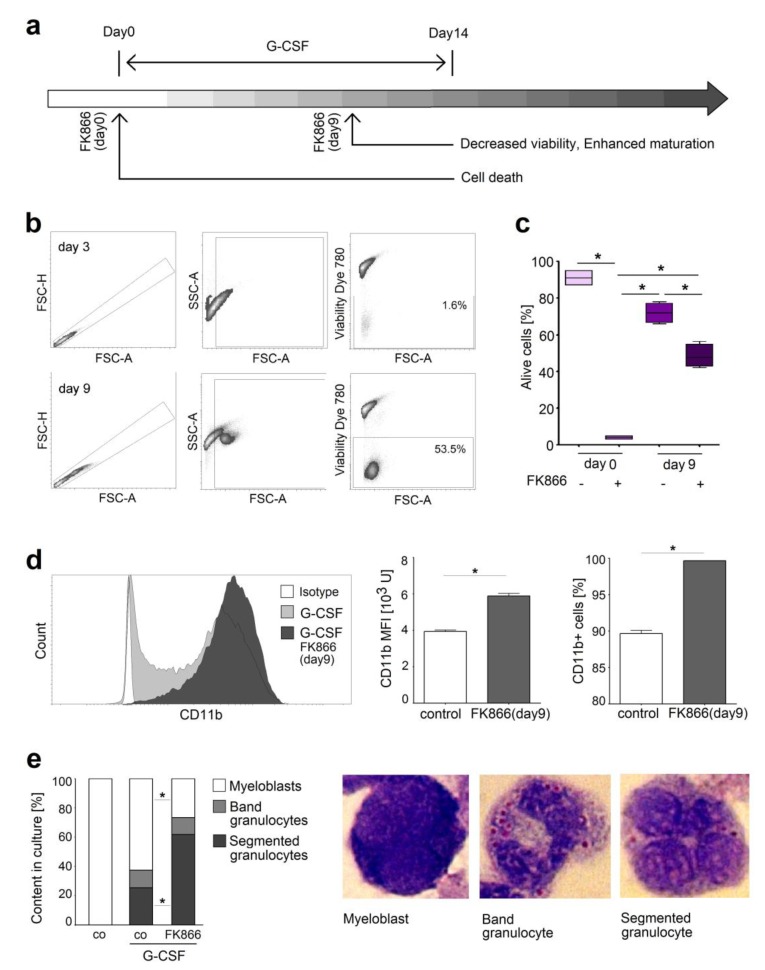
NAMPT inhibition increased apoptosis and induced maturation of the murine neutrophil-like cell line 32D. (**a**) The scheme of the experiment. Maturation of 32D cell line was induced with G-CSF, treatment with FK866 was performed at day 0 or 9; (**b**) Gating strategy for single alive cells. FK866 treatment at early stages of differentiation led to cell death; (**c**) FK866 treatment in early stages of 32D differentiation led to cell death, while at later stages moderately decreased viability; (**d**) FK866 treatment at late stages of 32D differentiation led to increased CD11b expression; (**e**) FK866 treatment at late stages of 32D differentiation was associated with a mature phenotype of the cells. Nuclear morphology was assessed after Giemsa staining using light microscope in at least 10 fields of view. *n* = 4 replications per group. Data are presented as median, minimal, and maximal values. * *p* < 0.05.

**Figure 6 ijms-20-05584-f006:**
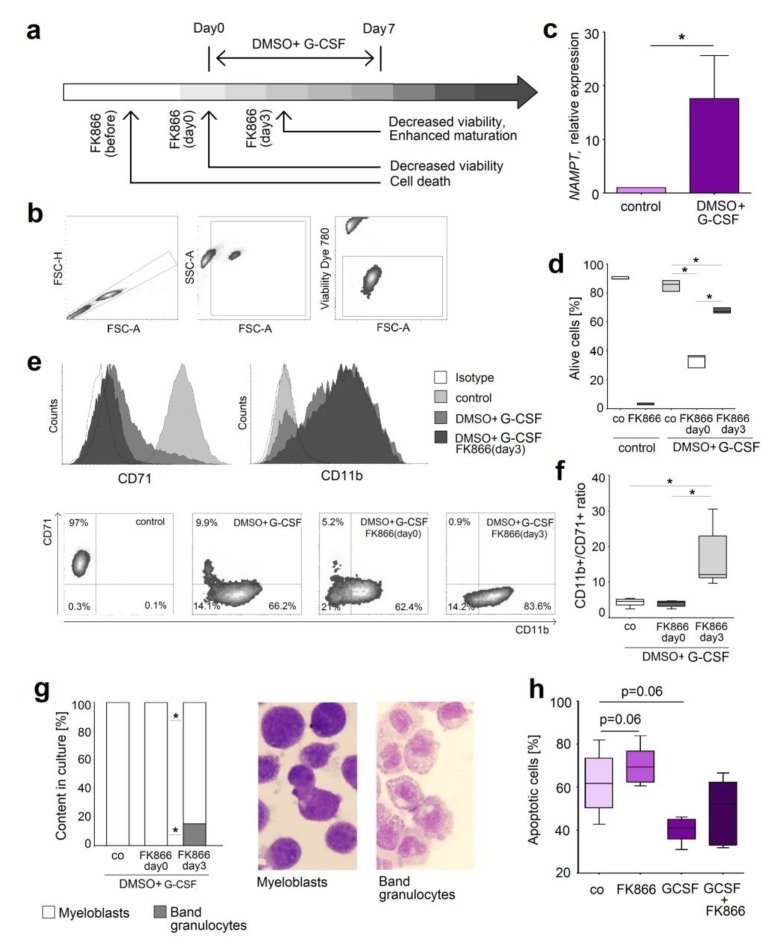
NAMPT inhibition increased apoptosis and induced maturation of the human neutrophil-like cell line HL-60. (**a**) The scheme of the experiment. Maturation of HL-60 cells into neutrophils was induced with DMSO and G-CSF, and treatment with FK866 was performed at different time points (at the beginning or 3 days after); (**b**) Gating strategy for single alive cells; (**c**) *NAMPT* gene expression is increased in HL-60 cells supplemented with DMSO and G-CSF in comparison to the control condition; (**d**) FK866 treatment at late stages of HL-60 differentiation led to moderately decreased viability; (**e,f**) FK866 treatment at late stages of HL-60 differentiation led to increased expression of maturation marker CD11b and decreased expression of marker of immature cells CD71; (**g**) FK866 treatment at late stages of HL-60 differentiation was associated with mature phenotype of the cells. Nuclear morphology was assessed after Giemsa staining using a light microscope in at least 10 fields of view; *n* = 4; (**h**) FK866 treatment decreases and G-CSF increases the viability of isolated neutrophils from healthy donors. *n* = 4. Data are presented as median, minimal and maximal values. * *p* < 0.05.

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
