# Peer review of "Neutrophil Maturation and Survival Is Controlled by IFN-Dependent Regulation of NAMPT Signaling"

_ijms, 2019, doi:10.3390/ijms20225584_

Round 1

Reviewer 1 Report

The manuscript “Neutrophil maturation and survival is controlled by IFN-dependent regulation of NAMPT signaling” by Siakaeva E et al. describes a INF-dependent modulation of NAMPT signaling pathway in the development, proliferation and survival of pre-, immature and mature neutrophils.The presented study shows interesting results, but there are some comments:

-In the figure 3a, the authors show that IFN-deficiency is associated with decreased survival of early progenitors and increased survival of mature neutrophils isolated from bone marrow of mice. It is not clear in the text and in the figure legend, what type of assay they performed to assess the percentage of dead and apoptotic cells. Furthermore, when they indicate in the graphs legends “apoptotic” and “dead by apoptosis”, do these expression mean early and late apoptosis respectively?

-In the figure 3b the authors show FACS analysis of negative and Annexin V positive population of pre-neutrophils and mature neutrophils obtained from WT and Ifnar1-/-mice. They describe the results of such experiments as an increase of apoptosis in pre-neutrophils and a decrease of that in the mature neutrophils, however, the total percentage of positive stained Annexin V cells in mature neutrohilps obtained from Ifnar1-/ mice appear similar to that observed for mature neutrohilps obtained from WT mice.

-In figure 3C, the authors analysed the relative expression of caspase 3 in pre-, immature and mature neutrophils obtained from WT and Ifnar1-/ mice, but they are not able to demonstrate if these results refer to active form of caspase 3 or not.

-In the present manuscript the authors also underlie a key role of NAMPT in the activation of tumor associated neutrophils. In this respect, in order to demonstrate a direct association between NAMPT gene expression and improving of neutrophils survival in tumors they should also test the effect of NAMPT inhibitor on modulation of apoptosis in tumor infiltrating neutrophils.

-In figure 4 b and c, the authors showed that NAMPT inhibition results in a reduction of survival of early progenitors from bone marrow of Ifnar1-/ mice, but is not clear what type of survival markers they tested in these experiments.

Reviewer 2 Report

General comments:

In this paper, Siakaeva et al report neutrophil maturation and survival is controlled by IFN-dependent regulation of NAMPT signaling by using murine and human model systems. The findings are highly interesting and provide important information for the role of NAMPT in granulopoiesis.

Minor revision may be required.

1) Fig. 4c: Symbols indicating the statistical significance may be missing for WT (alive cells and apoptotic cells).

2) Line 420: The final concentration (%) of DMSO used for induction of neutrophilic differentiation should be clarified.

Line 421 and Line 445: G-CSF (1000 U/ml) and G-CSF (10 ng/ml) are described. The same description is preferable.

Author Response

Reviewer 2

Comments and Suggestions for Authors
General comments:
In this paper, Siakaeva et al report neutrophil maturation and survival is controlled by IFN-dependent regulation of NAMPT signaling by using murine and human model systems. The findings are highly interesting and provide important information for the role of NAMPT in granulopoiesis. Minor revision may be required.

Thank you for all comments and remarks that improved our paper. Thank you for your time. We have marked all changes in the manuscript in red

1.Fig. 4c: Symbols indicating the statistical significance may be missing for WT (alive cells and apoptotic cells).

We thank the reviewer for this comment. To analyze the differences in alive, apoptotic and dead by apoptosis cells with or without FK866 treatment in different time points we used Mann-Whitney U-test for two independent samples. This method did not show us significant differences neither in alive cells from WT mice, treated or not treated with FK at day 5 (p=0,1) nor in apoptotic cells in the same conditions (p=0,2). Nevertheless, we observe the tendency of the impaired survival of cells from WT mice after FK866 treatment.

2. Line 420: The final concentration (%) of DMSO used for induction of neutrophilic differentiation should be clarified.
Line 421 and Line 445: G-CSF (1000 U/ml) and G-CSF (10 ng/ml) are described. The same description is preferable.

Thank you for these comments. We have corrected it accordingly and added this information to the Materials and Methods chapter.

Round 2

Reviewer 1 Report

The authors well addressed each point of the previous review!